# Activation of the Nrf2 Pathway Prevents Mitochondrial Dysfunction Induced by Caspase-3 Cleaved Tau: Implications for Alzheimer’s Disease

**DOI:** 10.3390/antiox11030515

**Published:** 2022-03-08

**Authors:** Francisca Villavicencio-Tejo, Margrethe A. Olesen, Alejandra Aránguiz, Rodrigo A. Quintanilla

**Affiliations:** Laboratory of Neurodegenerative Diseases, Instituto de Ciencias Biomédicas, Facultad de Ciencias de la Salud, Universidad Autónoma de Chile, Santiago 8910060, Chile; franvilla.tejo@gmail.com (F.V.-T.); molesen.2@gmail.com (M.A.O.); alejandracruzat411@gmail.com (A.A.)

**Keywords:** Alzheimer’s disease, tau, mitochondrial dysfunction, Nrf2, sulforaphane, antioxidant, neuroprotection

## Abstract

Alzheimer’s disease (AD) is characterized by memory and cognitive impairment, accompanied by the accumulation of extracellular deposits of amyloid β-peptide (Aβ) and the presence of neurofibrillary tangles (NFTs) composed of pathological forms of tau protein. Mitochondrial dysfunction and oxidative stress are also critical elements for AD development. We previously showed that the presence of caspase-3 cleaved tau, a relevant pathological form of tau in AD, induced mitochondrial dysfunction and oxidative damage in different neuronal models. Recent studies demonstrated that the nuclear factor (erythroid-derived 2)-like 2 (Nrf2) plays a significant role in the antioxidant response promoting neuroprotection. Here, we studied the effects of Nrf2 activation using sulforaphane (SFN) against mitochondrial injury induced by caspase-3 cleaved tau. We used immortalized cortical neurons to evaluate mitochondrial bioenergetics and ROS levels in control and SFN-treated cells. Expression of caspase-3 cleaved tau induced mitochondrial fragmentation, depolarization, ATP loss, and increased ROS levels. Treatment with SFN for 24 h significantly prevented these mitochondrial abnormalities, and reduced ROS levels. Analysis of Western blots and rt-PCR studies showed that SFN treatment increased the expression of several Nrf2-related antioxidants genes in caspase-3 cleaved tau cells. These results indicate a potential role of the Nrf2 pathway in preventing mitochondrial dysfunction induced by pathological forms of tau in AD.

## 1. Introduction

AD is the most prevalent neurodegenerative disease worldwide. AD is a progressive disorder that causes cognitive defects and behavioral changes, where the earliest symptoms are memory impairment that progresses gradually into severe dementia [1]. AD is characterized by the accumulation of Aβ and NFTs composed of pathological forms of tau, including hyperphosphorylated and truncated types [2,3]. Tau is a microtubule-associated protein that contributes to axonal transport, synaptic function, and microtubule stability. The cleavage of tau at D421 by caspase-3 (truncation) is an important contributor to NFTs formation and neurodegenerative changes observed in AD [4,5]. In particular, tau truncation by caspase-3 (D421) is present in AD brains [6], and increasing levels of this tau form have been detected in aged mice contributing to the deficits in synaptic plasticity and cognitive impairment [7]. 

Thus, nowadays, AD is considered a multifactorial disease in which mitochondrial dysfunction actively contributes to AD onset and progression. These mitochondrial abnormalities include bioenergetics defects, mitochondrial respiratory impairment, oxidative damage, and defects in mitochondrial dynamics that contribute to synaptic dysfunction observed in AD [8,9,10]. Importantly, we have demonstrated that the expression of caspase-3 cleaved tau in neuronal cell lines and primary neurons negatively affects mitochondrial health in response to calcium stress and Aβ treatment [9,11]. Furthermore, mitochondrial impairment induced by pathological modifications of tau can also affect neuronal communication [12,13,14]. Moreover, the expression of caspase-3 cleaved tau affected mitochondrial health-inducing depolarization and mitochondrial fragmentation by impairing the mitochondrial fusion regulator Opa1 [9,12,15]. Finally, the expression of this cleaved tau form reduced mitochondrial transport in hippocampal neurons, suggesting that these actions could be relevant to the synaptic deficiency observed in AD [12,13].

Since mitochondria are vulnerable to oxidative damage induced by an uncontrolled ROS increase, endogenous antioxidant defense systems could play an essential role in cell survival under physiological and pathological conditions [16]. Nrf2 is a redox-sensitive transcription factor that maintains redox homeostasis by regulating antioxidant-response element (ARE)-dependent transcription and the expression of antioxidant defense enzymes [17]. Since Nrf2 activates the antioxidant systems in response to oxidative stress, Nrf2 signaling is one of the primary systems counteracting mitochondrial-derived ROS production [18]. Nrf2-dependent antioxidant strategies include the increase of mitochondrial thiol redox systems, such as the mitochondrial glutathione (GSH) system and thioredoxin (TRX) [19,20,21]. Mitochondrial GSH is made in the cytosol, and then transported to the mitochondrial matrix by GSH peroxidases 1 and 4, glutathione-S-transferases (GSTs), and glutaredoxin-2 to prevent the increase of ROS [21]. The mitochondrial TRX system consists of TRX2 and TRX reductase 2 (TRXR2), which maintains TRX2 in a reduced state by using mitochondrial NADPH as a substrate [22]. In addition, TRX2 maintains the activities of the peroxidase Peroxiredoxin 3 and methionine sulfoxide reductases, while also directly reducing protein disulfides [23,24].

More interestingly, studies in a drosophila model of human tauopathy (Tau R406W) showed a reduction in TRX and mitochondrial superoxide dismutase 2 (SOD2) expression, which promoted Tau-induced neurodegeneration and neuronal apoptosis [25]. Complementary studies showed that AD brains presented a decrease in neuronal TRX1 and glutathione (*GRX*) expression [26]. Furthermore, Aβ neurotoxicity might be mediated by the oxidation of TRX-1 and subsequent activation of the apoptosis signal-regulating kinase-1 (ASK1) cascade [26]. These studies suggest that the deregulation of the GRX and TRX antioxidants system may contribute to AD pathogenesis [26].

Several reports have demonstrated the use of sulforaphane (SFN), an organic isothiocyanate compound found in cruciferous vegetables, including cauliflower, brussels sprouts, broccoli, white cabbage, and red cabbage, as a potent activator of the Nrf2 pathway in the SNC [27,28,29]. SFN, through Nrf2 pathway activation, can modulate diverse pathways in neuronal cells [29,30,31]. For example, SFN showed increasing tau [32] and p-tau [32,33,34] degradation. These effects against pathological forms of tau can also be observed in 3×Tg-AD mice [32], as well as cortical neurons treated with Aβ oligomers [32,33]. 

In the present study, we investigated the effect of Nrf2 activation and its downstream genes on mitochondrial dysfunction induced by the presence of caspase-3 cleaved tau in cortical neurons. Treatment with SFN prevented mitochondrial abnormalities and ROS increase induced by the expression of caspase-3 cleaved tau. Furthermore, these changes were accompanied by an increase in the Nrf2-dependent antioxidant genes, suggesting this pathway’s potential role in the prevention of mitochondrial failure induced by truncated tau. Therefore, Nrf2 could be considered a prominent player in the protection of the structural and functional integrity of the mitochondria in the brain.

## 2. Materials and Methods

### 2.1. Cell Culture 

Conditionally immortalized cortical neurons (CN1.4) [10,13] were cultured in 1×Dulbecco’s modified Eagle’s medium with high glucose (DMEM) (Mediatech CellGro, Corning, NY, USA), supplemented with 5% inactivated fetal bovine serum (Mediatech Inc., Manassas, VA, USA) with 1% penicillin/streptomycin (Mediatech Inc., CellGro, Corning, NY, USA), and incubated at 33° C and 5% CO_2_. In addition, these cells were transiently transfected with plasmids containing GFP, GFP-full-length tau (GFP-T4), and GFP-caspase-3 cleaved tau (GFP-T4C3), and after 48 h, transfected cells were treated with 10 μm of SFN (ChemCruz, Santa Cruz Biotechnology, Dallas, TX, USA) for 24 h. 

### 2.2. Tau Constructs 

Tau constructs tagged with GFP, GFP-full-length tau (GFP-T4), and GFP-caspase-3 cleaved tau (GFP-T4C3) were generated as previously described [9,11,13]. In addition, CN1.4 cells were transiently transfected with plasmids containing tau constructs using Lipofectamine 2000 (Thermo Fisher Scientific, Waltham, MA, USA) diluted in OptiMEM (Thermo Fisher Scientific, Waltham, MA, USA) [9,10,12]. Cells media were changed 24 h post-transfection, and analyses were conducted 48 h post-transfection. GFP and tau expression was verified with live-cell imaging, observing a 40% transfection efficiency in CN 1.4. Moreover, GFP, GFP-T4, and GFP-T4C3 expression levels were estimated by detecting GFP expression using Western blot.

### 2.3. Determination of Mitochondrial Length and Membrane Potential in Live Cells 

The mitochondrial length was evaluated in CN 1.4 cells double transfected by GFP and GFP-tau (s) forms and Mito-mCherry construct [9,11]. The mitochondrial length was obtained by measuring the lengths of individual mitochondria present in fluorescence images, which were obtained using high-resolution fluorescence microscopy (Leica, Wetzlar, Germany). We analyzed the mitochondrial population of 40–60 cells, measuring 25 images for each experiment using ImageJ software for analysis [9,11,12,13].

The mitochondrial membrane potential (MMP) levels were evaluated using the mitochondrial Mitotracker Red 2XRosH_2_ (MitoRed) dye in CN 1.4 cells [12,13]. Transfected cells were loaded with MitoRed in Krebs-Ringer-HEPES (KRH) buffer (136 mM NaCl, 20 mM HEPES, 4.7 mM KCl, 1.5 mM MgSO4, 1.25 mM CaCl2, 5 mM glucose; pH = 7.4) for 35 min, and then fluorescence images were taken. Fluorescence intensity was captured using a fluorescence microscope LX6000 (Leica, Wetzlar, Germany) under a 63×oil objective. We analyzed 30–40 cells on average for the quantification process, measuring at least 25 images for each experiment. Mitochondrial potential levels are presented as the pseudo ratio (∆*F*/*F*0), where *F* is the average of the fluorescence signal (*F*) per area in every image, and (*F*0) represents the intensity of background fluorescence [11,12,13]. 

### 2.4. Determination of ROS Levels 

ROS levels were evaluated using CellRox^TM^ dye (Molecular Probes, Thermo Fisher Scientific) [24]. Transfected cultured cells were incubated with 0.5 μM CellRox in KRH–glucose buffer at 37 °C for 30 min [13]. Fluorescence images were acquired using the same exposure time and gain to minimize dye photo-bleaching [35,36]. CellRox fluorescence (*F*) was determined in each cell, and each image’s background (*F*0) was subtracted before normalization per cell area. Normalization was performed to ameliorate differences in the dye distribution inside the cells that present differences in cellular morphology. Fluorescence images were acquired using a fluorescence microscope (Leica LX6000, Wetzlar, Germany). 

### 2.5. Western Blot Analysis

CN1.4 cells were lysed in Triton lysis buffer, including a protease inhibitor cocktail (Roche Applied Science, Mannheim, Germany) and a phosphatase inhibitor (ThermoFisher, Waltham, MA, USA) [10]. Total protein extracts (30 μg) were separated on a sodium dodecyl sulfate (SDS)-polyacrylamide gel, and subsequently transferred onto PVDF membranes. After blocking in 5% non-fat milk and 0.1% Tween-20 in phosphate-buffered saline (PBS), the membranes were incubated with rabbit monoclonal anti-Nrf2 (1:1000; Cell Signaling, Danvers, MA, USA), mouse monoclonal Catalase (1:1000; Santa Cruz, Dallas, TX, USA), mouse monoclonal HO-1 (1:1000; Santa Cruz, Dallas, TX, USA), and mouse monoclonal NQO-1 (1:1000; Santa Cruz, Dallas, TX, USA) antibodies. To test the equal protein loading, the membranes were subsequently re-tested with anti-actin (1:3000; Santa Cruz, Dallas, TX, USA) antibody, as indicated. Primary antibody signal was determined by horseradish peroxidase (HRP)-linked goat anti-mouse or anti-rabbit secondary antibodies (1:3000; Thermo Fisher, Waltham, MA, USA), and protein expressions were detected using enhanced chemiluminescence (ECL, Thermo Fisher, Waltham, MA, USA). Expression levels of the indicated proteins were estimated related to the intensity of housekeeping protein signals using ImageJ software (NIH, Bethesda, MD, USA). 

### 2.6. Real-Time Polymerase Chain Reaction 

The expression of Nrf2 (NM_010902.5), Keap1 (NM_001110307.1, HO-1 (NM_010442.2), NQO-1 (NM_008706.5), GR1 (NM_010344.4), TRXr1 (NM_001042523.1), and GCS (NM_010295.2) mRNA levels were analyzed by real-time polymerase chain reaction (RT- PCR) (Table 1) [37]. Total RNA was isolated from 100 mg of CN 1.4 lysates transfected with GFP, GFP-T4, and GFP-T4C3 using the Trizol reagent (Life Technologies, Thermo Fisher Scientific), according to the manufacturer’s protocol. RNA yield and purity were measured in a microplate reader (TECAN, Infinite 200 PRO series. One μg of total RNA was subjected to reverse transcription using the ImProm-II Reverse Transcription System (Promega), following the manufacturer’s protocol. The cDNA was stored at −20 °C for further use. Each cDNA sample was diluted 10 times with nuclease-free water for qPCR analysis. The real-time PCR reaction was performed in triplicate in the Aria Max (Agilent technologies) in a final volume of 10 μL [13]. Amplification conditions consisted of an initial denaturation at 95 °C for 10 min, followed by amplification of 40 cycles (95 °C for 15 s, 60 °C for the 20 s, and 72 °C for 20 s). A melting curve analysis was performed immediately after amplification from 55 to 95 °C. Values were normalized to 18S expression levels using the ΔCT method. The following table presents the primers sequence of genes evaluated:

### 2.7. Measurement of ATP Concentration 

ATP concentration was measured in cortical neurons transfected with GFP, GFP-T4, and GFP-T4C3 in the total lysates using a luciferin/luciferase bioluminescence assay kit (ATP determination kit #A22066, Molecular Probes, Invitrogen) [38]. The amount of ATP in each sample was calculated from standard curves and normalized to the total protein concentration.

### 2.8. Statistical Analysis 

The data are expressed as the mean ± standard error (SEM), with at least three experiments indicated in the corresponding figures. All samples included in these studies were analyzed to normality distribution using the Kolmogorov–Smirnov test [37]. Later, the obtained data were analyzed using Student’s *t*-test with Dunnett’s post hoc test or, if analyzing more than two groups, ANOVA followed by Bonferroni’s post hoc test. *p* < 0.05 and *p* < 0.001 were considered statistically significant. All statistical analyses were performed using Prism software (GraphPad Software Inc., San Diego, CA, USA). 

## 3. Results

### 3.1. Activation of the Nrf2 Pathway Prevent Mitochondrial Dysfunction Induced by Caspase-3 Cleaved Tau

Immortalized cortical neurons were co-transfected with Mito-mCherry to examine mitochondrial morphology in situ, as well as mitochondrial membrane potential levels (Figure 1A–D). We previously showed that truncated tau expression reduced mitochondrial length and mitochondrial membrane potential levels (MMP) [10,11]. Here, we observed that GFP-T4C3 presented a reduced mitochondrial length than GFP and GFP-T4 cells (Figure 1A,B). GFP and GFP-T4 present a uniform mitochondrial morphology distributed throughout the whole cell body (Figure 1B). However, GFP-T4C3 cells showed more rounded and fragmented mitochondria (Figure 1B). Quantification analysis shows that GFP-T4C3 cells present a decrease of two-fold in mitochondrial length, compared with GFP or GFP-T4 cells (Figure 1C). Interestingly, when GFP-T4C3 cells were treated for 24 h with 10 μM of SFN, the mitochondrial fragmentation was prevented entirely, reaching similar values of mitochondrial length of GFP and GFP-T4 cells (Figure 1B,D). For mitochondrial membrane potential determinations, GFP and GFP-tau (s) transfected cells were loaded with MitoRed, and treated with 0.5 μM thapsigargin (25 min) to induce a transient cytosolic calcium increase to test mitochondrial function [11,39]. Therefore, MitoRed fluorescence intensity levels were obtained from live-cell images taken in a fluorescence microscope (Leica, Germany). Mitochondrial potential levels were severely affected in GFP-T4C3 cells, reaching a more than four-fold decrease compared with GFP or GFP-T4 transfected cells. More importantly, GFP-T4C3 treated with SFN showed prevention of mitochondrial depolarization reaching similar mitochondrial potential levels to GFP and GFP-T4 cells (Figure 1D). These studies suggest that SFN prevents mitochondrial impairment induced by caspase-3 cleaved tau.

### 3.2. Activation of the Nrf2 Pathway Improves Mitochondrial Health by Rescuing ATP Production Loss Induced by Caspase-3 Cleaved Tau

Due to their high activity and signaling, neuronal cells require a constant energy supply, which makes them particularly vulnerable to mitochondrial dysfunction [40]. Decreased ATP production is a common feature in neurodegenerative diseases, such as AD, and can be caused by various mechanisms, including the impaired activity of mitochondrial respiratory complexes, alterations in glucose uptake, glycolysis, and the TCA cycle or uncoupling [41]. To further understand how the presence of caspase-3 cleaved tau affected mitochondrial functions, we measured ATP production in immortalized cortical neurons that expressed GFP, GFP-T4, and GFP-T4C3 (Figure 2). The expression of GFP-T4C3 presented a significant decrease in basal ATP levels compared to GFP and GFP-T4 expressing cells (Figure 2A). Furthermore, we treated transfected cells with 0.5 μM of thapsigargin, and we observed that ATP production was markedly decreased, reaching similar values in all these three conditions (Figure 2A). Significantly, when cells were treated for 24 h with 10 μM SFN, the loss in ATP production was restored in GFP-T4C3 cells treated with thapsigargin (Figure 2A).

### 3.3. Sulforaphane Prevented ROS Increase Induced by Caspase-3 Cleaved Tau Expression

Complementary to mitochondrial function determinations, we analyzed ROS production levels in cells transfected with GFP, GFP-T4, and GFP-T4C3 using Cell Rox dye (Figure 3). We observed that the expression of caspase-3 cleaved tau significantly increased basal ROS levels compared with GFP and GFP-T4 cells (Figure 3A). Moreover, transfected cells treated with thapsigargin (0.5 μM, 30 min) showed an increase in ROS levels; however, under this condition, GFP-T4C3 cells showed a more significant increase in Cell Rox signal compared to GFP and GFP-T4 cells (Figure 3B). Interestingly, ROS increase in GFP-T4C3 cells showed a fluorescence pattern similar to mitochondrial localization, indicating an essential role of this organelle in this effect (Figure 3A). Significantly, SFN completely reduced ROS-increased levels in cells expressing caspase-3 cleaved tau (GFP-T4C3), compared with ROS levels shown in control or full-length tau cells (Figure 3B). These studies suggest that incubation with SFN prevented the excess ROS production present in GFP-T4C3 cells, facilitating the protection of mitochondrial function, as observed by the normalization of ROS levels.

### 3.4. Treatment with SFN Induces Activation of the Nrf2 Pathway in Immortalized Cortical Neurons

To verify the activation of the Nrf2 pathway by SFN, we performed rt-PCR and analyzed the mRNA expression of central antioxidant response genes activated by Nrf2 in GFP, GFP-T4, and GFP-T4C3 cells (Figure 4). Under basal conditions, the mRNA levels of Nrf2 in the GFP condition and the presence of caspase-3 cleaved tau remained relatively similar, except for GFP-T4 cells, where lower levels of mRNA expression were observed (Figure 4). As a part of this Nrf2 activation mechanism, SFN modifies Keap1 cysteine residues, functioning as sensors for numerous oxidants and electrophiles, leading to Nrf2 activation [18]. Notably, when the cells were treated with SFN (10 μM, 24 h), Nrf2 expression was significantly increased in all three conditions (Figure 4A). To further analyze the Nrf2 pathway, we evaluated the mRNA levels of the KEAP1, the negative regulator of the Nrf2 pathway (Figure 4B). When cells were exposed to SFN, a mild increase in the KEAP1 expression in GFP and GFP-T4 cells was shown. Moreover, caspase-3 cleaved tau cells exposed to SFN presented KEAP1 mRNA levels similar to the control condition.

Furthermore, we analyze the expression of primary Nrf2-ARE dependent genes, including NQO1, HO-1, Glutamate-cysteine ligase (GCS), glutathione-disulfide reductase (GR1), and thioredoxin reductase 1 (TRxR 1), [26] (Figure 4C–F). The analysis of NQO1, a cytosolic flavoprotein that catalyzes quinine detoxification, showed that its basal expression was diminished in GFP, GFP-T4, and GFP-T4C3 expressing cells (Figure 4C). On the other hand, the treatment with SFN induced a marked increase in this gene expression, reaching an increase of more than three-fold in the GFP and GFP-T4C3 expressing cells (Figure 4C). Although, this increase was lower in GFP-T4 cells than in control, maintaining a similar tendency in the expression of Nrf2. In addition, we evaluated HO-1 levels (Figure 4D), showing that SFN treatment significantly increased its expression in the three cell conditions (Figure 4D). Further analysis of GCS and GR1 mRNA levels showed that SFN treatment increased GCS and GR1 levels in GFP, GFP-T4, and GFP-T4C3 cells (Figure 4E,F). Finally, we evaluated the levels of the mammalian selenoprotein, thioredoxin reductase 1 (TRxR 1) (Figure 4G). Intriguingly, GFP-T4C3 cells showed a higher basal expression in TRxR 1 levels than GFP and GFP-T4 cells. Moreover, SFN treatment increased TRxR1 in all conditions indicated. However, this increase was higher in cells that expressed GFP-T4C3 (Figure 4G).

These results indicate that SFN induces the expression of antioxidant genes dependent on Nrf2 activity in immortalized cortical neurons. These events could be related to the prevention of ROS increase and mitochondrial impairment induced by the treatment with SFN in GFP-T4C3 cells. 

### 3.5. Nrf2 Pathway Activation Increase the Antioxidant Protein Expression 

Complementarily, we checked protein expression levels of several antioxidant proteins regulated by the Nrf2 pathway [42,43]. As we previously observed for mRNA levels, under basal conditions, Nrf2 protein levels remained similar in the GFP and GFP-T4 cells, while a lower decrease in Nrf2 expression was observed in GFP-T4C3 cells. After the treatment with SFN (10 μm, 24 h), Nrf2 expression was significantly increased in all cell conditions indicated (control, full-length, and caspase-3 cleaved tau). NQO-1 expression in cortical neurons showed similar results, with lower levels in the control conditions and higher expression when they were treated with SFN. Moreover, we measured catalase levels, whose expression is also related to Nrf2 activity [42]. Catalase is one of the crucial antioxidant enzymes that mitigates oxidative stress to a considerable extent by converting hydrogen peroxide to produce water and oxygen [42,43]. Under control conditions, GFP, full-length, and caspase-3 cleaved tau expressing cells showed relatively similar protein expression, and SFN treatment significantly increased catalase levels in all three conditions. 

## 4. Discussion

This study aimed to elucidate the possible role of the Nrf2 pathway on mitochondrial impairment induced by caspase-3 cleaved in neuronal cells. Nrf2 plays a central role in the cytoprotective response to oxidative stress and is critical for maintaining mitochondrial redox homeostasis [44]. Here, we reported that, in cells expressing truncated tau, mitochondria presented a fragmented morphology and higher levels of ROS and mitochondrial depolarization [11]. These observations indicate that truncated tau impairs mitochondrial function, which likely contributes to the neuronal dysfunction observed in AD. Interestingly, SFN treatment, a natural activator of Nrf2, prevented mitochondrial fragmentation in caspase-3 cleaved tau expressing cells (Figure 1). In addition, mitochondrial depolarization presented in GFP-T4C3 cells was prevented by the treatment with SFN (Figure 1). Moreover, the treatment with SFN for 24 h prevented ROS production in truncated tau-expressing cells, suggesting that Nrf2 activation improves mitochondrial health. 

Related to our findings, Kovac and colleagues showed that mitochondrial–ROS production is increased in brain tissue of Nrf2-KO mice [45]. They also demonstrated that Nrf2 impacts cellular bioenergetics by controlling substrate availability and the efficiency of mitochondrial fatty acid oxidation, thus affecting oxidative phosphorylation and increasing mitochondrial-ROS production [45]. Here, we observed that caspase-3 cleaved tau induces a decrease in ATP production, which correlates with mitochondrial depolarization shown in truncated tau expressing cells (Figure 2). In the same context, Holmström et al. observed an impaired activity of complex I of the respiratory chain under conditions of Nrf2 deficiency in neuronal cells of Nrf2-knockout mice [46]. Furthermore, Nrf2 deficiency leads to mitochondrial depolarization, reduced ATP levels, and impaired respiration [46]. On the other hand, other studies showed that basal MMP was increased when Nrf2 was genetically activated by knockout or knockdown of Keap1 protein, indicating that Nrf2 could regulate mitochondrial activity. Furthermore, the Nrf2 pathway contributes to mitochondrial function by modulating NADH and FADH2, both of which are important substrates for mitochondrial respiration [47,48]. 

Interesting studies have suggested an essential role of the Nrf2 pathway in promoting mitochondrial integrity under oxidative conditions [49]. For example, the isolated brain of rats that were subjected to a single dose of the Nrf2 activator SFN showed resistance to opening the mitochondrial permeability transition pore (mPTP) caused by the oxidant tert-butyl hydroperoxide [49]. The formation and opening of the mPTP is a fundamental factor in mitochondrial dysfunction, and this action can induce mitochondrial depolarization, decrease ATP production, release mitochondrial contents, and lead to cell death [50,51,52,53,54,55,56]. Interestingly, some studies suggest an important action of the Nrf2 pathway modulating mPTP opening induced by oxidative stress [57,58]. For example, Greco and coworkers observed a slight inhibition of peroxide-induced mPTP opening in mitochondria isolated from the brains of normal rats injected with SFN [49].

Furthermore, additional studies from Greco et al. demonstrated that SFN administration to rats dramatically inhibits redox-regulated mPTP opening by liver mitochondria, and increases immunoreactive levels of mitochondrial antioxidant-related proteins [49]. More importantly, we previously described that the use of mPTP inhibitor cyclosporin A (CsA) was influential in preventing mitochondrial fragmentation, decrease in MMP, and mitochondrial calcium handling defects showed in neuronal cells that expressed caspase-3 cleaved tau [11]. However, further studies are needed to elucidate if the prevention of mitochondrial dysfunction induced by Nrf2 is produced by mPTP regulation in caspase-3 cleaved tau expressing cells.

Emerging evidence demonstrates that the accumulation of phosphorylated and oligomerized tau is a toxic event that affects neuronal function [59,60]. In this context, evidence obtained by Jo et al. showed that activating the Nrf2 pathway with SFN reduced the abnormal accumulation of hyperphosphorylated tau by inducing the expression of autophagy adapter protein NDP52 in neurons [36]. The NDP52 protein presents three ARE in its promoter region, and its expression is strongly induced by Nrf2 activation, facilitating the elimination of hyperphosphorylated tau to the autophagic degradative pathway [36,61]. Although we did not observe these effects in caspase-3 cleaved tau cells treated with SFN (Figure 5), we cannot rule out the possibility that this pathway could be contributing to preventing mitochondrial dysfunction induced by truncated tau.

Finally, our results showed that treating cortical neurons with SFN increased the Nrf2-ARE-dependent genes expression in all conditions indicated (Figure 4). Interestingly, we detected a significant increase in mRNA TRX1 expression in caspase-3 cleaved tau cells, compared with GFP-T4 cells (Figure 4). TRX1 is a critical enzyme that regulates several processes involved in decreasing oxidative stress-reducing intracellular disulfides and upholding the redox homeostasis within the cell [62]. Previous findings have shown that TRX1 levels were elevated in the cerebrospinal fluid of AD patients [63]. Moreover, the immunohistochemical studies of TRX1 revealed that only cytosolic localization was observed in the hippocampus CA1 of AD patients, whereas TRX1 in the control patients was observed in the nucleus, with no difference in the expressing levels between control and AD patients [63]. These observations suggest that the translocation from the nucleus to the cytosol may diminish this reduction and, ultimately, the function of critical signaling molecules and transcription factors regulated by TRX1 [64]. Notably, under the effect of SFN, NQO-1, GCS, and GR1 levels were elevated in full-length and caspase-3 cleaved tau expressing cells, indicating activation of the Nrf2-ARE pathway.

Furthermore, previous findings reported a significant increase in HO-1 expression in the post-mortem brain of AD temporal cortex and hippocampus compared to aged-matched control [65]. Moreover, an increase in NQO1 activity and expression was found in astrocytes and neurons of the AD brain [66]. Additionally, studies in aged APP/PS1 AD mouse models showed a reduction in Nrf2, NQO1, GCL catalytic subunit (GCLC), and GCL modifier subunit (GCLM) mRNA levels [67]. These studies indicate that Nrf2-related genes contribute to neurodegeneration in AD, and they may be responsible for mitochondrial impairment observed during this disease. 

Our findings presented here indicate the importance of the Nrf2 pathway in reducing mitochondrial and oxidative damage induced by pathological forms of tau. Furthermore, it is essential to understand the mechanism underlying the protection induced by Nrf2-dependent genes, and to elucidate other protective effects on mitochondrial function and the consequences of this protection on neuronal dysfunction in AD. 

## Figures and Tables

**Figure 1 antioxidants-11-00515-f001:**
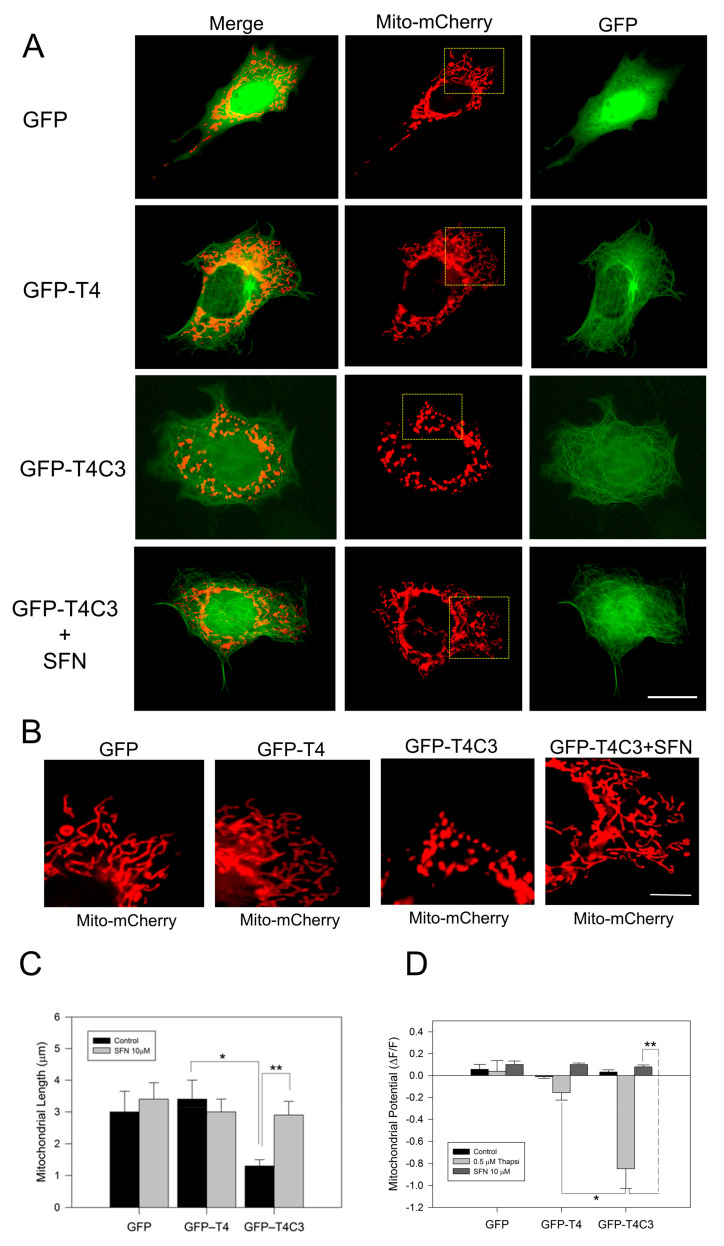
Activation of the Nrf2 pathway prevents the mitochondrial failure induced by truncated tau. (**A**) CN1.4 cells were co-transfected with Mito-mCherry and GFP, GFP-T4, and GFP-T4C3 tau forms to evaluate mitochondrial length. Alternatively, GFP, GFP-T4, and GFP-T4C3 were treated with Sulforaphane (SFN, 10 μM 24 h) to evaluate mitochondrial length and mitochondrial membrane potential levels. (**B**) Magnification of mitochondria morphology from Mito-mCherry presented in **A**. Representative fluorescent images show that treatment with SFN prevented mitochondrial fragmentation induced by caspase-3 cleaved tau in immortalized cortical neurons. (**C**) Quantification of mitochondrial length obtained from fluorescent images obtained from double transfected CN 1.4 cells. (**D**) Determination of mitochondrial membrane potential levels. SFN treatment prevented mitochondrial depolarization induced by truncated tau. Data are presented as Mean ± SE, *n* = 5. Statistics differences were calculated by the t-Student test. * *p* < 0.001, ** *p* < 0.05. A, Bar = 20 μm; B Bar = 5 μm.

**Figure 2 antioxidants-11-00515-f002:**
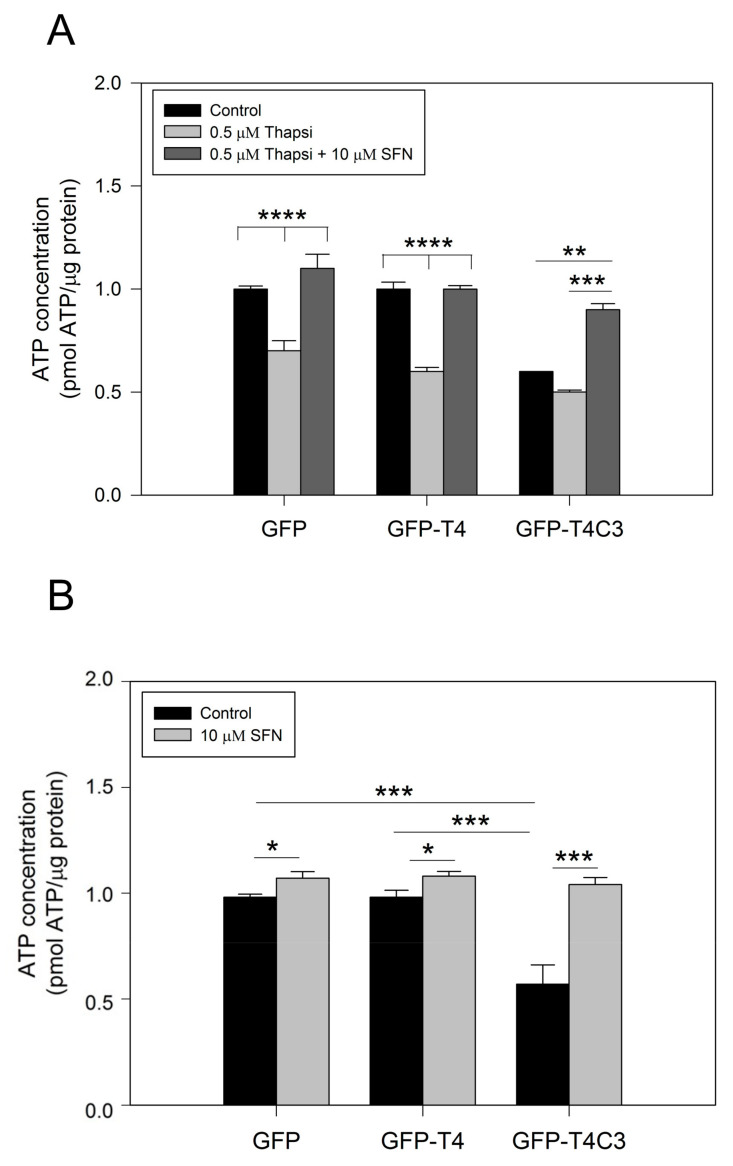
Treatment with sulforaphane prevents ATP loss induced by truncated tau. (**A**) CN 1.4 cells were transfected with GFP and GFP-tau(s) forms (full-length and truncated) and were treated with thapsigargin (0.5 μM, 1 h) to determine ATP levels. Treatment with Sulforaphane (SFN, 10 μM 24 h) prevented ATP loss induced by caspase-3 cleaved tau. (**B**) ATP levels determination in transfected cells exposed to SFN (10 μM, 24 h) and control conditions. Graphs bars represent Mean ± SE, *n* = 4. The statistical differences were calculated by a one-way ANOVA test. * *p* < 0.5, ** *p* < 0.01, *** *p* < 0.0007, **** *p* < 0.0001.

**Figure 3 antioxidants-11-00515-f003:**
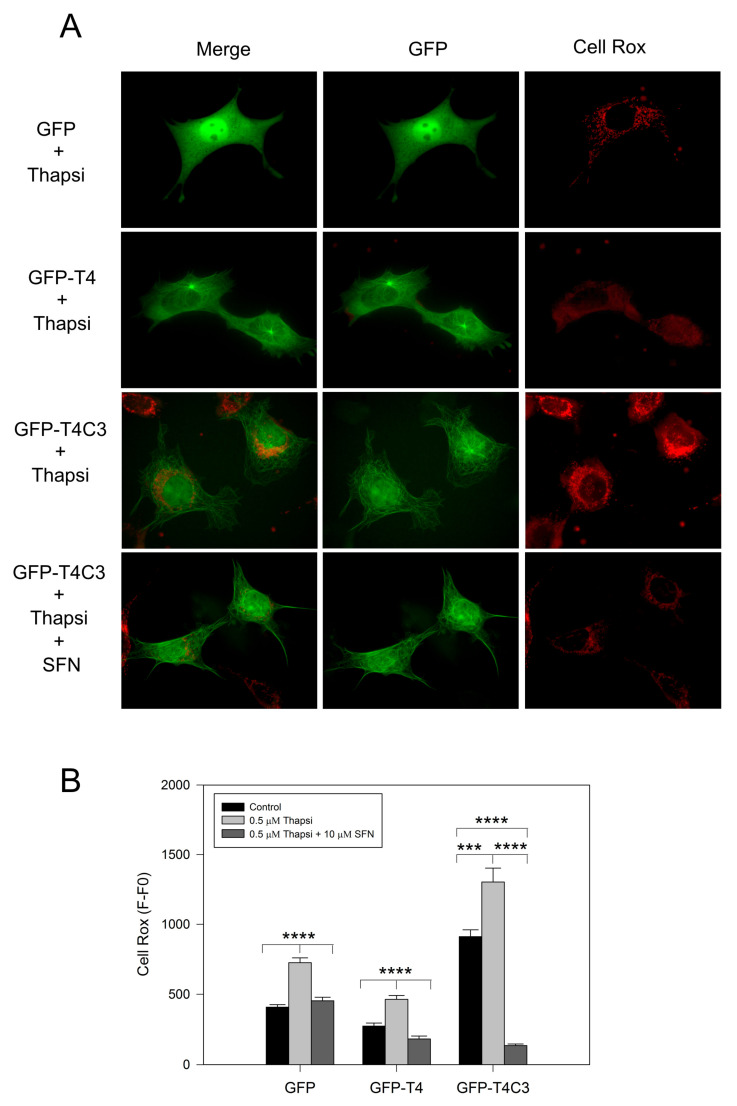
Activation of the Nrf2 pathway decreased ROS production in cortical neurons that express caspase-3 cleaved tau. (**A**) Fluorescence representative images of cells transfected with GFP, GFP-T4, and GFP-T4C3 tau forms were treated with thapsigargin (0.5 μM, 1 h) and Sulforaphane (SFN, 10 μM 24 h). ROS production was measured using Cell Rox dye (see Materials and Methods section). (**B**) Cells expressing caspase-3 cleaved tau showed excess in ROS production compared to full-length tau. SFN treatment prevented this ROS increase from reaching similar values to those observed in cells expressing full-length tau. Data are mean ± SE, *n* = 4. *p* < 0.05 indicates differences between groups calculated by the one-way ANOVA test. *** *p <* 0.0005, **** *p* < 0.0001. Bar = 20 μm.

**Figure 4 antioxidants-11-00515-f004:**
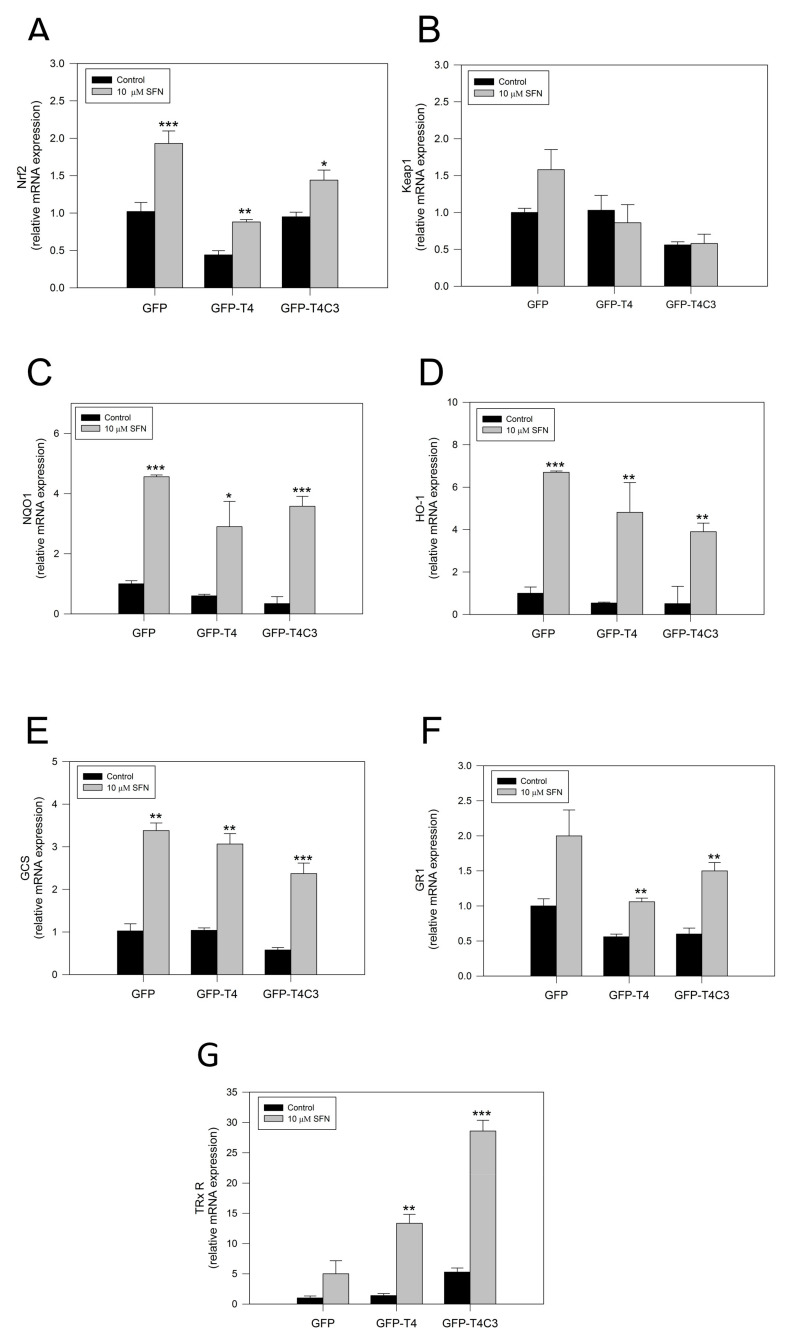
Sulforaphane induces the activation of the Nrf2 pathway in immortalized cortical neurons. Relative mRNA expression of (**A**) Nrf2, (**B**) Keap1, (**C**) NQO1, (**D**) HO-1, (**E**) GCS, (**F**) GR1, and (**G**) TRxR in CN1.4 cells transfected with GFP, GFP-T4, and GFP-T4C3 tau forms. Treatment with SFN (10 μM, 24 h) increased antioxidant gene expression. Data are presented as the Mean ± SE, *n* = 4. Differences were calculated by one-way ANOVA tests. * *p* < 0.5, ** *p* < 0.01, *** *p* < 0.001.

**Figure 5 antioxidants-11-00515-f005:**
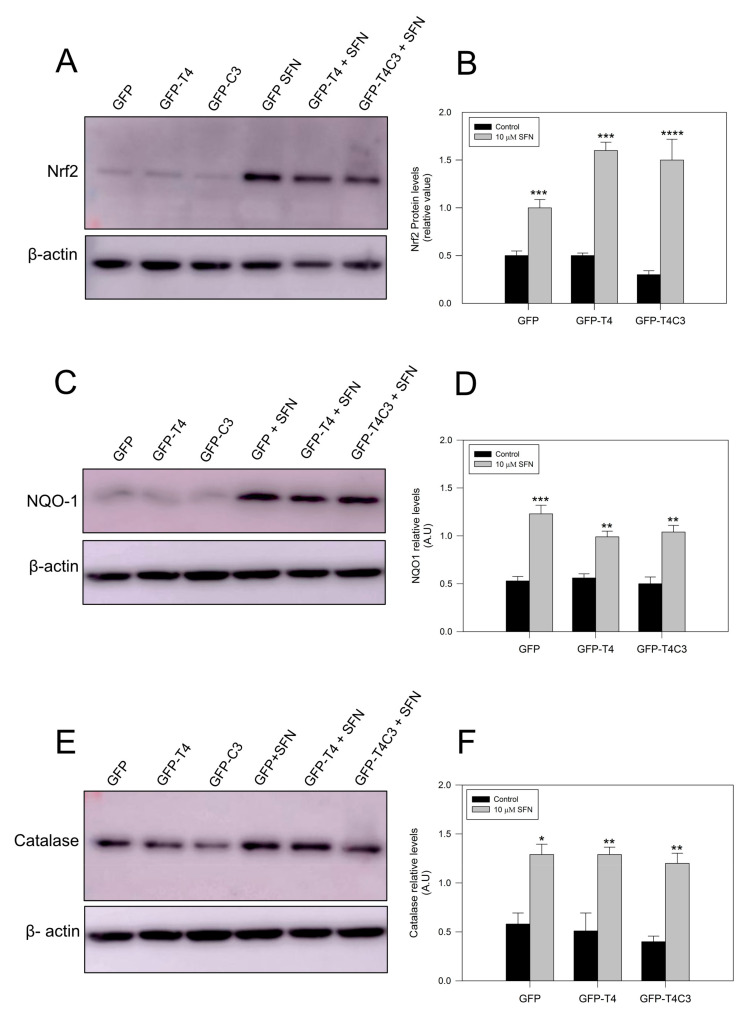
Treatment with sulforaphane increases the antioxidants protein expression in immortalized cortical neurons. (**A**,**C**,**E**) are representative western blot images of immortalized cortical neurons transfected with GFP, GFP-T4, and GFP-T4C3, showing the protein expression of Nrf2, NQO-1, and Catalase. Treatment with SFN (10 μM 24 h) increased the expression of these proteins in all conditions indicated. (**B**,**D**,**F**) showed densitometry analyses for Nrf2, NQO-1, and Catalase expression. Data are presented as the Mean ± SE, *n* = 4. Statistical analysis were performed using one-way ANOVA test. * *p* < 0.01, ** *p* < 0.0061, *** *p* < 0.0003, **** *p* < 0.0001.

**Table 1 antioxidants-11-00515-t001:** Primers sequences of Nrf2/ARE representative genes evaluated.

Gen	Forward Primer	Reverse Primer
Nrf2	5′-GCT TTT GGC AGA GAC ATT CCC-3′	5′-CTG CCA AAC TTG CTC CAT GTC-3′
Keap1	5′-GAT GGC CAC ATC TAC GCA GT-3′	5′-GCG GAG TTA AGC CGG TTA GT-3′
HO-1	5′-TGA CAC CTG AGG TCA AGC AC-3′	5′-ATC TTG CAC CAG GCT AGC AG-3′
GR1	5′-CCA CGG CTA TGC AAC ATT CG-3′	5′-GAT CTG GCT CTC GTG AGG AA-3′
NQO-1	5′-CTG CCA TGT ACG ACA ACG GT-3′	5′-ATC GGC CAG AGA ATG ACG TT-3′
TrxR1	5′-AGT CAC ATC GGC TCG CTG AAC T-3′	5′-GAT GAG GAA CCG CTC TGC TGA A-3′
GCS	5′-GGG GTG ACG AGG TGG AGT A-3′	5′-GTT GGG GTT TGT CCT CTC CC-3′
GAPDH	5′-CAT CAC TGC CAC CCA GAC TG-3′	5′-ATG CCA GTG AGC TTC CCG TTC AG-3′

## Data Availability

Data is contained within the article.

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
