# Peer review of "Activation of the Nrf2 Pathway Prevents Mitochondrial Dysfunction Induced by Caspase-3 Cleaved Tau: Implications for Alzheimer’s Disease"

_antioxidants, 2022, doi:10.3390/antiox11030515_

Round 1

Reviewer 1 Report

The authors are undoubtedly competent in the topic of research, and the presented work seems to be a continuation of the work of the whole group for some time now. The methods of research are well presented in the article and interesting results are obtained, which are discussed in a special section involving research works on this topic.

Comments:

Could the authors please specify what is the concrete originality of the research?

On the subject of Alzheimer's disease, I want to mention the works where it was found that 30% of patients diagnosed with Alzheimer's disease neurofibrillary tangles consisting of hyperphosphorylated Tau were not detected. (DOI: 10.1212/01.wnl.0000118212.41542.e7; DOI: 10.1097/00005072-199311000-00012; DOI: 10.1002/ana.410300206). Why did you use сonditionally immortalized cortical neurons (CN1.4) specifically?

The authors emphasize in the titles of the results based on literature data that sulforaphane is a potent activator of the Nrf2 path-80 way in the SNC:

3.1. Activation of the Nrf2 pathway prevent mitochondrial dysfunction induced by caspase-3 cleaved tau

3.2. Activation of the Nrf2 pathway improves mitochondrial health by rescuing ATP production loss induced by caspase-3 cleaved tau

However, the actual activation of the Nrf2 pathway is shown only in section 3.4. Maybe section 3.4 should be listed first.

Line 280 When cells were exposed to SFN showed an increase in the KEAP1 expression in GFP and GFP-T4 cells. Here unreliable increase in the KEAP1 expression in GFP and GFP-T4 cells?

The authors quantified catalase but missed other important antioxidant enzymes: superoxide dismutase and glutathione peroxidase. Why didn't they do a Western blot of these enzymes?

Line 12 Probably a mistype NTFS-should be NTFs.

Fig. 1C The figure legends are poorly visible.

Line 212 with Sulforaphane (SFN, 10 m 24 h). A typo in the indicated concentration.

Fig. 2 not quite understandable ATP units.

Line 496 move the reference to the new line.

Section 4.5 has another control, GFP-C3. Not previously mentioned or described. Is this a typo?

Line 326 GFP-C3. Is this a typo?

Line 331 Is this the beginning of the Discussion section?

Reviewer 2 Report

Thank you for the opportunity to review manuscript entitled ‘Activation of the Nrf2 pathway prevents mitochondrial dysfunction induced by caspase-3 cleaved tau. Implications for Alzheimer's disease’ by Francisca Villavicencio-Tejo and co-authors.

In the present study, authors investigated the role of the Nrf2 pathway in preventing mitochondrial dysfunction induced by pathological forms of tau in AD. The manuscript is well written and logical and presents data appropriate for publication in Antioxidants journal. However, there are some key points that require clarification.

  1. Figure 5B does not correspond to figure5A. The intensity of the luminescence of beta actin is much greater than your diagrams indicate. The ratio of target protein / control should be others. Check these calculations and provide all images of blots that you have included in the calculations.
  2. The ‘bar' is not displayed on the figure 1A.
  3. Mitochondria in Figure 1 should be magnified to confirm changes in their shape and size.
  4. Line 219: ** < 0,05 replace with p <0.05. This remark also applies to Figure 2 (lines 242-243), Figure 3 (line 267), Figure 4 (line 308), and Figure 5 (line 330) captions.
  5. Please, replace the ‘,’ with ‘.’ in the y-axis in the figure1C. Apply this to all figures in your manuscript.
  6. Lines 242-243: ‘*p <0,05, **p <0,0021, ***p<0,0007’  replace with common notation ‘*p <05, **p <0.01, ***p<0,001’.  
